# Incidence of Recurrent Exertional Heat Stroke in a Warm-Weather Road Race

**DOI:** 10.3390/medicina56120720

**Published:** 2020-12-21

**Authors:** Rebecca L. Stearns, Yuri Hosokawa, William M. Adams, Luke N. Belval, Robert A. Huggins, John F. Jardine, Rachel K. Katch, Robert J. Davis, Douglas J. Casa

**Affiliations:** 1Department of Kinesiology, Korey Stringer Institute, University of Connecticut, Storrs, CT 06269, USA; yurihosokawa@waseda.jp (Y.H.); wmadams@uncg.edu (W.M.A.); lukebelval@texashealth.org (L.N.B.); Robert.huggins@uconn.edu (R.A.H.); johnjardinemd@gmail.com (J.F.J.); Rachel.k.katch@gmail.com (R.K.K.); douglas.casa@uconn.edu (D.J.C.); 2Faculty of Sport Sciences, Waseda University, Tokyo 169-8050, Japan; 3Department of Kinesiology, University of North Carolina at Greensboro, Greensboro, NC 27412, USA; 4Institute for Exercise and Environmental Medicine, Texas Health Presbyterian Hospital Dallas and University of Texas Southwestern Medical Center, Dallas, TX 75321, USA; 5Southcoast Health, New Bedford, MA 02740, USA; davisrj@southcoast.org

**Keywords:** recovery, cold-water immersion, cooling, mass medical care

## Abstract

*Background and Objectives:* Exertional heat stroke (EHS) survivors may be more susceptible to subsequent EHS; however, the occurrence of survivors with subsequent EHS episodes is limited. Therefore, the purpose of this study was to evaluate the incidence of participants with repeated EHS (EHS-2+) cases in a warm-weather road race across participation years compared to those who experienced 1 EHS (EHS-1). *Materials and Methods:* A retrospective observational case series design was utilized. Medical record data from 17-years at the Falmouth Road Race between 2003–2019 were examined for EHS cases. Incidence of EHS-2+ cases per race and average EHS cases per EHS-2+ participant were calculated (mean ± SD) and descriptive factors (rectal temperature (T_RE_), finish time (FT), Wet Bulb Globe Temperature (WBGT), age, race year) for each EHS was collected. *Results:* A total of 333 EHS patients from 174,853 finishers were identified. Sixteen EHS-2+ participants (11 males, 5 females, age = 39 ± 16 year) accounted for 11% of the total EHS cases (n = 37/333). EHS-2+ participants had an average of 2.3 EHS cases per person (range = 2–4) and had an incidence rate of 2.6 EHS per 10 races. EHS-2+ participants finished 93 races following initial EHS, with 72 of the races (77%) completed without EHS incident. Initial EHS T_RE_ was not statistically different than subsequent EHS initial T_RE_ (+0.3 ± 0.9 °C, *p* > 0.050). Initial EHS-2+ participant FT was not statistically different than subsequent EHS FT (−4.2 ± 7.0 min, *p* > 0.050). The years between first and second EHS was 3.6 ± 3.5 year (Mode: 1, Range: 1–12). Relative risk ratios revealed that EHS patients were at a significantly elevated risk for subsequent EHS episodes 2 years following their initial EHS (relative risk ratio = 3.32, *p* = 0.050); however, the risk from 3–5 years post initial EHS was not statistically elevated, though the relative risk ratio values remained above 1.26. *Conclusions:* These results demonstrate that 11% of all EHS cases at the Falmouth Road Race are EHS-2+ cases and that future risk for a second EHS episode at this race is most likely to occur within the first 2 years following the initial EHS incident. After this initial 2-year period, risk for another EHS episode is not significantly elevated. Future research should examine factors to explain individuals who are susceptible to multiple EHS cases, incidence at other races and corresponding prevention strategies both before and after initial EHS.

## 1. Introduction

Exertional heat stroke (EHS) is one of the top three causes of death during sport participation [1,2,3,4]. While it is clear that survival can be as high as 100% when aggressive whole body cold water immersion (CWI) is promptly applied [1,5], recovery following an EHS is much more variable, especially when CWI is not utilized within the 30 min following collapse [6,7,8,9,10,11]. The time course for successful and full recovery hinges on initial treatment [8], but may also be due to patient-specific risk factors, such as fitness, heat acclimatization and previous history of EHS. While previous history of EHS remains a cited risk factor for future EHS risk [1,9,12,13], very little clinical data exist to support this statement. The current literature relies heavily on military data and a few case studies of individuals with multiple instances of EHS [13,14,15,16,17,18,19]. Therefore, it has yet to be established, beyond a handful of case reports [14,17,18,20], that repeated EHS cases exist within the athlete or civilian population.

### Background

Currently, two theories are posited for why subsequent EHS risk exists: the former suggests that initial care may dictate outcome and future risk, while the latter suggests patients may possess an innate risk profile that may not be modifiable or managed. For the latter, emerging evidence suggests that risk mitigation for EHS recurrence may not always be possible due to genetic factors that have been shown to be associated with increased EHS risk [17,20,21,22,23]. Therefore, some EHS cases may be a manifestation of a genetic disorder (e.g., malignant hyperthermia) that increases an individual’s risk for EHS rather than a temporary impairment of heat tolerance. Certainly, the potential for the two to co-exist within an individual is possible [19]; however, this remains yet to be reported within the existing scientific literature.

The argument that future EHS risk increases due to lingering effects of an initial EHS episode, while in most scenarios is logical, remains a long-held and unsubstantiated claim. Largely, this is due to the inability to ascertain whether the thermoregulatory dysfunction observed following an EHS event is an outcome of the injury or an undefined, innate characteristic. Post-EHS thermoregulatory dysfunction can be explained by the irreversible or prolonged tissue and organ damage that ensues from delayed treatment or inferior cooling methods [5,7,9,11,19]. Additionally, prompt and aggressive cooling using CWI [1,5] followed by biochemical recovery monitoring (i.e., systemic inflammatory response and renal and hepatic function) and gradual reintroduction of exercise and environmental heat exposure has allowed for the successful return to sport and physical activity [14,24]. This suggests that (1) an initial EHS insult may only temporarily impair the body’s thermoregulatory system, (2) that CWI may mitigate this impairment, and (3) that recovery is possible when paired with correct on-site treatment, guided/gradual reintroduction of exercise, and heat acclimatization [14,18,24].

The inability to examine factors related to repeat EHS incidents with high level of evidence research designs (like randomized controlled studies) and the limited data available from case reports or military populations [13,15,25] leave medical providers with little research to guide medical decisions pertaining to future EHS risk. To understand if EHS patients are at risk for another EHS, it is critical to first understand if these occurrences exist within a robust dataset outside of military or case reports. Therefore, the purpose of this study was to describe and evaluate the incidence of multiple EHS episodes (EHS-2+) across participation years (2003–2019) at the Falmouth Road Race (FRR), compared to those who experienced 1 EHS episode (EHS-1) during the same period. Additionally, we sought to describe and examine the relationship of sex and Wet Bulb Globe Temperatures (WBGT) between EHS-2+ and EHS-1 groups. We hypothesized that EHS-2+ participants would represent a small fraction of the yearly EHS cases at the FRR medical tent, but that these individuals would demonstrate different patterns to their subsequent EHS episodes (such as differences in WBGT temperatures, finish times or overall race exposure/participation).

## 2. Materials and Methods

This retrospective observational case series examined medical tent records of runners treated for EHS at the FRR (11.3 km, Falmouth, Massachusetts, USA) over a 17-year period (2003–2019). The FRR was chosen due to its documented EHS rate, which is higher than any other reported road race (2 cases per 1000 finishers or 15–20 each year) [5]. Healthcare providers (e.g., physicians, nurses, and athletic trainers) skilled in the diagnosis and treatment of EHS identified and treated patients in the medical tent. The diagnostic criteria for EHS was an initial rectal temperature (T_RE_) greater than 40 °C with concurrent neuropsychological dysfunction (e.g., confusion, combativeness, collapse), which was assessed by a qualified healthcare professional at the medical tent.

Successful FRR finishes, identified as years where EHS patients completed the FRR and were not admitted to the medical tent, were included for comparison. Records of overall race finishers and finish times were also collected. This study was approved by the University’s Institutional Review Board and qualified as exempt from consent due to the use of previously collected de-identified medical records. Only medical information collected at the FRR’s medical tent was available and included for analysis. Participants who had only one instance of EHS and those who had more than one EHS at FRR were grouped accordingly. All records of EHS-2+ participants were paired with the patient’s EHS cases from multiple years. Data were provided by the road race medical team. It was prepared in a de-identified manner for analysis by the research team. Finish time (FT) and race participation data for non-EHS years were included when available; however, some cases were excluded because pairing or matching these data was not possible (e.g., if the name on medical chart did not match the race bib number or any registered runners for that race).

Meteorological data (e.g., dry bulb temperature, dewpoint temperature, and cloud cover) were collected from the nearest available weather observing station located at Otis Air National Guard Base (41.65° N, −70.52° W), which is approximately 18.8 km from the race start and operated in joint effort between the National Weather Service, the Federal Aviation Administration, and the Department of Defense. WBGT were not routinely recorded during race events and were, therefore, computed from meteorological data using the Heat Stress Adviser [26] (version 2005) software package [27]. Input data into this model include air temperature, dewpoint temperature, cloud cover, and time of day. The model was developed for warm season conditions (May–September) and was tested in a variety of geographic regions in the U.S., including Oklahoma, Texas, Minnesota, and New York. WBGT estimates are accurate to ±1.1 °C [26,27]. Mean WBGT from the start to two hours after the start of the race (generally 9:00 am to 11:00 am) were used to represent the environmental conditions experienced by the participating runners for each year.

All statistical analyses were performed utilizing SPSS (Version 26; IBM Corporation, Armonk, NY, USA) except where noted otherwise. Descriptive statistics included age, FT, sex, initial T_RE_ (upon admittance to the medical tent), subsequent EHS initial T_RE_, race WBGT data, and years between EHS cases. Paired samples T-tests were utilized to determine differences between first and second EHS for initial T_RE_, FT, and race WBGT. Hedge’s G effect size (ES) calculations with weighted pooled standard deviations [28] were used to assess differences in initial T_RE_ and FT between first and second EHS as well. These are reported with their respective 95% confidence intervals (CI) as (ES [95% CI]). An effect size < 0.2 was considered small, 0.2–0.8 was considered a medium effect and anything above 0.8 was considered a large effect. Linear mixed effects regression models utilizing FT as the outcome of interest were used to determine whether EHS-1 or EHS-2+ participants altered their race pace/time after their first EHS, while also accounting for age, sex and WBGT. Individual race ID was included as a random effect on intercepts to account for the repeated measurements of participants across years. This analysis was performed in R (Version 3.6.1; R Core Team, 2019) using the lme4 package (Version 1.1-21; Bates et al., 2015). These results are graphed using notched box plots. Pearson’s bivariate correlations were used for comparisons with weather data and EHS occurrence. Strength of the correlation coefficient was defined according to Chang et al. [29]. To compare EHS-1 vs. EHS-2+ participant WBGT correlation coefficients, a Fisher r-to-z transformation was performed, a value of z was applied to assess the significance of the difference between two correlation coefficients (two tailed test). For comparison of future risk following initial EHS, EHS-1 finishers were examined, except for data from the most recent race year (2019) due to the inability to describe future risk/exposures. Relative risk ratios [30] of EHS between EHS-1 and EHS-2+ groups were then calculated for each year following the initial EHS episode, spanning years 2 to 6. Only participants with data available for these each time frame (i.e., 4 years) were utilized. Therefore, a participant who had their initial EHS in 2017 could not be included in the 4-year relative risk ratio analyses (since the last year of data collection was 2019). Additionally, relative risk ratio 1 year post-EHS was not possible to calculate because it created a flawed 2 × 2 contingency table (since there is no chance that someone categorized as EHS-2+ would not have their second heat stroke in a 2-year time frame, this resulted in a 0 for box C). Incidence of EHS-2+ participants by successful finishers and average EHS cases were evaluated. Outside of capturing incidence rate, relative risk ratio and average EHS cases per person, the 3rd and 4th EHS episode results were not analyzed further due to the small sample size (n = 4). Alpha level was set a priori at or equal to 0.050, and trends were defined as *p* ≤ 0.100. Values are reported as mean ± SD.

## 3. Results

Of the 174,853 finishers and 333 EHS patients over the 17-year period, a total of 16 patients (11 males, five females, age at first EHS = 39 ± 16 years) were identified as EHS-2+. When calculated from total race finishers, EHS-2+ participants had an incidence rate of 0.21/1000 finishers. When EHS-2+ cases were removed from the general EHS and finisher data, the incidence rate of single episode EHS cases was 1.70/1000 finishers. Over the 17-year period, the 16 EHS-2+ participants represented 11% of the total EHS cases (n = 37/333 EHS cases) and had an average of 2.3 EHS cases per person (range = 2–4). This resulted in an average of 2 ± 1 EHS-2+ participants diagnosed and treated in the medical tent per year. When examining all race years, including EHS and non-EHS years, EHS-2+ participants completed 145 total races (average races per person = 9 ± 6, range = 2–17, mode = 6). The duration in years between first and second EHS was 3.6 ± 3.5 years, range = 1–12 years, mode = 1 year. For the 37 EHS-2+ cases in their cumulative 145 races, the overall incidence rate was 2.6 EHS-2+ cases per 10 races (26% of the races they completed resulted in treatment for EHS in the medical tent).

### 3.1. EHS-2+ Participant Comparisons of First and Second Initial Rectal Temperatures and Finish Times

Specific to the EHS-2+ cohort, the first EHS mean initial T_RE_ (41.5 ± 0.7 °C) was not significantly different than second EHS initial T_RE_ (41.2 ± 0.7 °C, *p* = 0.199) (Figure 1). The mean difference between the first EHS FT (52.0 ± 9.5 min, n = 15) and the second EHS FT (57.7 ± 11.8 min, n = 14); was 4.2 ± 7.0 min or 7.7 ± 11.6% slower (*p* = 0.056, Figure 2). Some FT data for first EHS or second EHS (n = 3) were not available because participants did not officially finish the race. ES calculations revealed a moderate effect of T_RE_ (−0.49 [−3.73, 2.74]) and small effect of FT (0.36 [−0.58, 1.30]).

Following the initial EHS race year, EHS-2+ participants completed 93 subsequent races, successfully completing 72 of the 93 races (77%) without another EHS episode. Per person, this represented an average of 6 ± 5 successful races following the initial EHS year. These data only represent 12 patients, as 4 of the patients never returned to the race following back-to-back years with diagnosed EHS episodes (Figure 3). Interestingly, females had their second EHS significantly sooner than males (Females: 1.6 ± 0.5 years after first EHS, range: 1–2 years vs. Males: 4.5 ± 4.0 years after first EHS, range: 1–12 years, *p* = 0.039).

### 3.2. Finish Time Results for EHS-1 and EHS-2+ Participants

As some EHS-1 participant data could not be paired with FT from other years, those data have been excluded from the following analyses (n = 65). The remaining 268 EHS-1 participants and their FT from EHS years and successful finish years (n = 736 observations) were compared to determine any change in FT after their initial EHS. These analyses accounted for the effects of sex, age and WBGT. EHS-1 participant’s FT prior to and following their EHS were similar (*p* > 0.050) for females (65.6 ± 15.1 min vs. 62.2 ± 14.3 min), and males (57.6 ± 10.7 min vs. 57.1 ± 13.4 min). See Figure 4 for pooled data and Figure 5 for individual data.

The same analyses were repeated to determine if EHS-2+ participant’s FT were different prior first EHS compared to races after the first EHS but prior to and including the second EHS. These analyses also accounted for age and WBGT at the time of the race. We found that of the 16 participants and 140 observations, FT was similar (*p* > 0.05). When FT was analyzed by sex, finish time decreased after first EHS for females (65.6 ± 15.1 min vs. 62.2 ± 14.3 min), but males demonstrated similar FT results (57.6 ± 10.7 min vs. 57.1 ± 13.4 min). See Figure 6. Notches on the boxplots suggest that FT are not statistically significantly different for males, but that FT is significantly different for females (at alpha = 0.050).

### 3.3. Relative Risk Ratios for Future EHS (EHS-1 vs. EHS-2+ Participants)

For comparison of future risk following initial EHS, EHS-1 finishers (n = 296), representing 16 years of data, were included in a relative risk ratio analysis with EHS-2+ participant data. Data from the most recent race year (2019) were subsequently excluded from these analyses due to the inability to describe future risk/exposures. Additionally, some EHS-1 participants’ data could not be linked to previous or future race participation; therefore, 236 of the initial 296 EHS-1 participants were included for these analyses, representing 579 race finishes. Relative risk ratios of EHS between EHS-1 and EHS-2+ participants were then calculated for each year following the initial EHS episode, spanning years 2 to 6. Only participants with complete data available for these each time frame (i.e., 4 years following initial EHS) was utilized. As data available for each time span varied, the total races observed (EHS races and non EHS races) are listed here as: years post first EHS (EHS-1 races, EHS-2+ races, total races in the model). Year 2 (sample = 374, sample = 21, sample = 395), Year 3 (sample = 384, sample = 25, sample = 409), Year 4 (sample = 399, sample = 42, sample = 441), Year 5 (sample = 354, sample = 33, sample = 387), Year 6 (sample = 324, sample = 49, sample = 373).

Figure 7 depicts the relative risk ratio of having a second EHS in reference to how many years have passed since the first EHS at this road race. It is important to note that 56% of EHS-2+ participants had their second EHS event within the first 2 years and 81% within 4 years.

Results from Figure 7 revealed a significantly elevated risk (relative risk ratio = 3.33, *p* = 0.050) for a second EHS at the 2 year time point only. While relative risk ratios remained high for years 3–6 (relative risk ratios = 2.00, 1.60, 1.77, 1.26 respectfully), statistically, it was not significant.

### 3.4. Weather and EHS Outcomes

Weather values for estimated maximum WBGT was significantly, and fairly correlated with number of EHS-2+ cases each year (r^2^ = 0.239, *p* = 0.046). First EHS estimated maximum race WBGT for EHS-2+ cases was higher (25.1 °C ± 3.0 °C), though not significant compared to subsequent EHS episode estimated maximum race WBGT (23.9 ± 2.0 °C, *p* = 0.301). EHS-1 cases were also significantly and very strongly correlated with WBGT (r^2^ = 0.703, *p* < 0.001); however, when compared to EHS-2+ participants, WBGT correlation coefficients were not statistically different (*p* = 0.072)—see Figure 8.

## 4. Discussion

This is the first dataset to demonstrate the existence of EHS survivors who suffer subsequent EHS episodes and provides the first incidence rate associated with these cases. The largest finding from this study is that 11% of the yearly EHS cases observed are participants who have had a previous EHS episode at this same race. While this race has previously reported an EHS incidence rate of 2.13/1000 participants [5], this study expands upon previous work and demonstrated an EHS-2+ participant incidence rate of 0.21/1000 finishers. This study revealed that patients who suffer multiple EHS episodes at this road race are at highest risk for a subsequent EHS within the first 2 years following their initial EHS. While subsequent EHS risk demonstrated a gradual decrease from years 2–6, these analyses did not account for other risk factors associated with EHS [1,4,9]; therefore, these relative risk ratios may increase or decrease based on the presence or absence of individual heat tolerance risk factors.

There is some, though limited, research to compare these results to. The most relevant is a study by the French military, which reported that 15% of military patients described a prior EHS episode [15]. How this conveys to the current dataset is limited as this was an unconfirmed, self-reported EHS diagnosis, and the timing in relationship to their current EHS was not reported. When considering biochemical recovery, recent data have demonstrated elevated markers up to 16 days post-EHS based on aggregated consensus of normal values [31]. Additionally, an altered heat shock response has been noted at 30 days post-EHS in a mouse model [32], though neither study examined future exposure or the occurrence of future EHS episodes.

When examining race specific factors (T_RE_ and FT), EHS-2+ participants had similar T_RE_ and FT compared to their first EHS. Additional FT linear mixed model analyses demonstrated that EHS-1 had similar FT before and after their initial EHS. However, EHS-2+ female participants ran races significantly faster after their initial EHS, and improved their time by an average of just over 3 min. Females also had a significantly shorter time frame between first and second EHS (females: 1.6 ± 0.5 years, vs. males: 4.5 ± 4.0 years after first EHS, *p* = 0.039). The faster FT and shorter time frame between first and second EHS, while interesting, is speculative at this point and may be a focus for future research.

Estimated WBGT values between first and second EHS were similar (*p* > 0.050); however, as with previous literature linking WBGT to EHS rates at this race [33], we found a significant and fair correlation with number of EHS-2+ patients at the medical tent and estimated WBGT (r^2^ = 0.2391, *p* = 0.046). Previously, WBGT has explained 48–63% of variance observed in EHS patients at FRR [34]; however, these data do not separate out those with repeated EHS events. When analyzed separately, estimated maximum WBGT explained 70% of the variance observed in EHS-1, whereas estimated maximum WBGT only explained 24% of the variance in EHS-2+ participants. While the EHS-1 and EHS-2+ group WBGT coefficients were not statistically different (*p* = 0.072), it was defined as a trend (*p* < 0.100). This warrants further research and clinical considerations in order to determine if WBGT may not be as influential in EHS-2+ cases, and if so, what other factors are better predictors. If risk factors between EHS-1 and EHS-2+ groups predict EHS at different magnitudes, it would be of utmost clinical importance to define these differences for the purpose of guiding recovery from EHS and educating EHS victims on future EHS risk factors.

While EHS history for these patients outside of this race is unknown, previous EHS history at this road race is associated with an increased risk for having another EHS episode; however, this risk is diminished after 2 years. It is important to note that 56% of EHS-2+ participants had their second EHS event within the first 2 years and 81% within 4 years. Additionally, the ability to return following an EHS episode was highly variable in this population. For example, four participants had two sequential years with EHS and never returned to the race, while two individuals participated every year with 10 or more years between their EHS episodes.

One of the newly emerging arguments for risk of future EHS relates to the potential for a genetic or inherent factor [20]. One of the gaps in this theory is the fact that there are very few reported cases of repeated instances of EHS episodes, regardless of a known genetic or transient risk factor. At a minimum, these data could demonstrate that a small sample of individuals are susceptible to repeated EHS episodes, and this calls for further examination of the factors associated with their susceptibility. Preventing future EHS cases will ultimately hinge on determining whether they truly possess an inherent risk factor for EHS or if an initial EHS episode (in combination with the treatment provided and recovery in the months following) has increased their risk of suffering another EHS.

One limitation of this dataset is that any EHS incidences outside of this race and many factors known for impacting immediate heat tolerance and performance, such as training history/fitness, sickness, hydration status, and heat acclimatization status are not accounted for. Additionally, each participant’s recovery and return to activity following their initial EHS is unknown. However, despite lacking this information, the data were able to demonstrate a significant risk of repeated EHS within the first 2 years following an initial EHS episode. As these data only capture this specific race, exposure and incidence of multiple EHS events only stands to increase, since it is likely that this race is not the only event these runners participate in each year. Therefore, these data should be considered a conservative snapshot of these instances.

## 5. Conclusions

These data are the first to report repeated EHS cases (EHS-2+) in an athlete or civilian population and is the largest single dataset to analyze athletes with multiple EHS events. Given that the variance of EHS-2+ case incidence explained by WBGT reduced from EHS-1 to EHS-2+ participants despite no difference in post- race T_RE_ or FT, underlining mechanisms of EHS-2+ cases may be different between the initial and subsequent EHS episodes. Future research designed to elucidate both inherent and acquired EHS risk factors present within and outside of this race is needed to investigate if initial and subsequent EHS indeed follow a different mechanism of injury.

Nevertheless, medical providers at mass medical tents and those providing return to activity medical oversight, should acknowledge that there is a possibility for repeated EHS cases from year to year, with the highest risk occurring in the first two years following the first EHS episode. Following an initial EHS episode, medical providers are encouraged to conduct a thorough medical history and assess what unique factors the runner may possess that makes them vulnerable to EHS. Pre-participation exam screening should also be most sensitive to a reported EHS event if it has occurred in the last two years. Lastly, because this dataset suggests that patients who suffer multiple EHS episodes are at the highest risk for a subsequent EHS within the first 2 years following their initial EHS, additional and more conservative precautions during this time frame may be warranted.

## Figures and Tables

**Figure 1 medicina-56-00720-f001:**
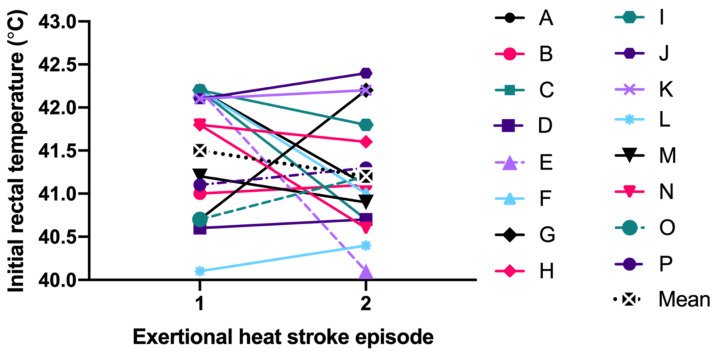
Comparison of initial rectal temperature between first and second exertional heat stroke episode.

**Figure 2 medicina-56-00720-f002:**
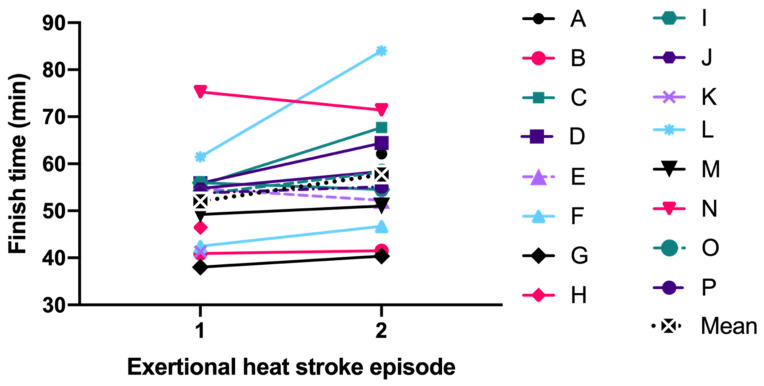
Comparison of finish time between first and second exertional heat stroke episode.

**Figure 3 medicina-56-00720-f003:**
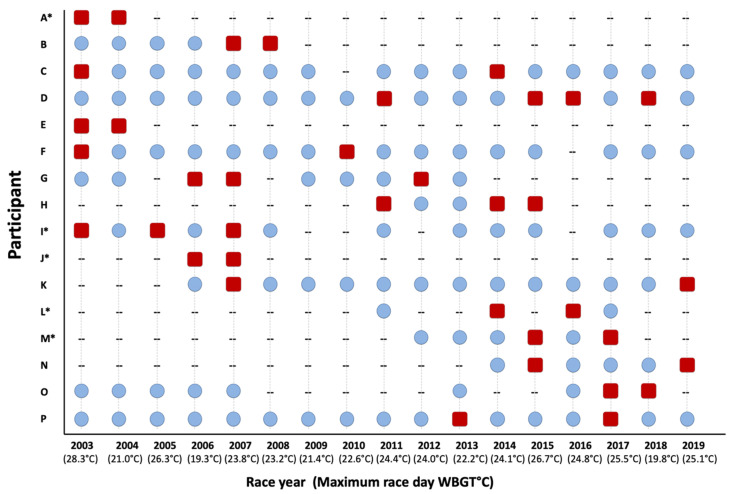
Individual participant depiction of successful road race participation years (blue circles), race participation years with an exertional heat stroke incident (red squares) and non-participation years (--), * next to participant letter indicates female. WBGT = wet bulb globe temperature.

**Figure 4 medicina-56-00720-f004:**
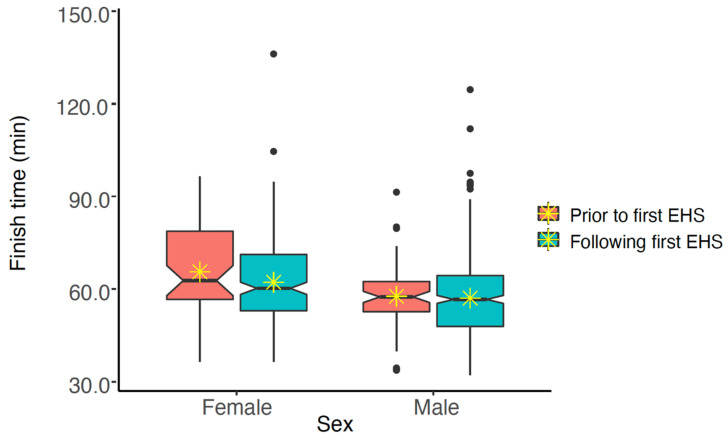
Notched box plot of finish times for males and females prior to and following exertional heat stroke (EHS) at the Falmouth Road Race for those with only having 1 EHS at the race. The box represents the interquartile range, spanning the 25 to 75 percentiles of the data. * indicates mean of the data. The line shows the median of the data, and the ends of each notch represent the 95% confidence interval of the median. Notches on the boxplots overlap, suggesting that finish times are not statistically significantly different (at alpha = 0.050).

**Figure 5 medicina-56-00720-f005:**
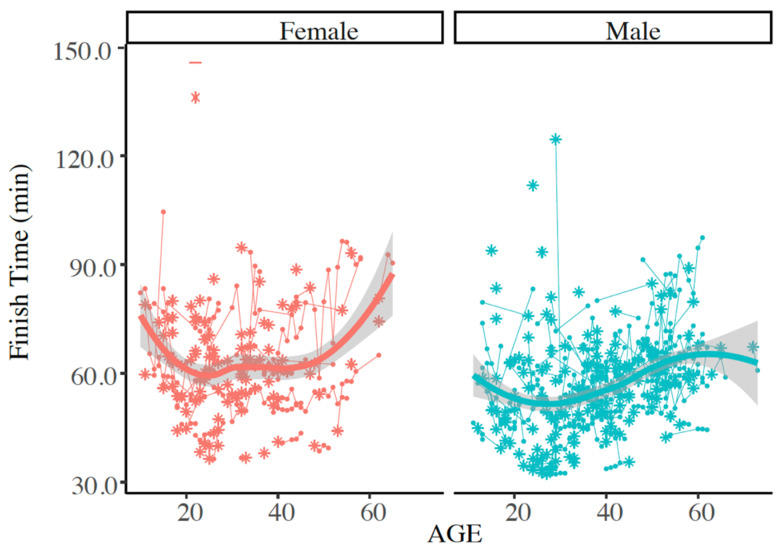
Individual finish times for race year with exertional heat stroke (EHS) as indicated by the asterisk (*), and any subsequent successful Falmouth Road Race finishes as shown by this trajectory plot. Plot represents those with only 1 documented EHS at the race. Solid line indicates mean of finish times. Finish times were not statistically different between males and females at (at alpha = 0.050).

**Figure 6 medicina-56-00720-f006:**
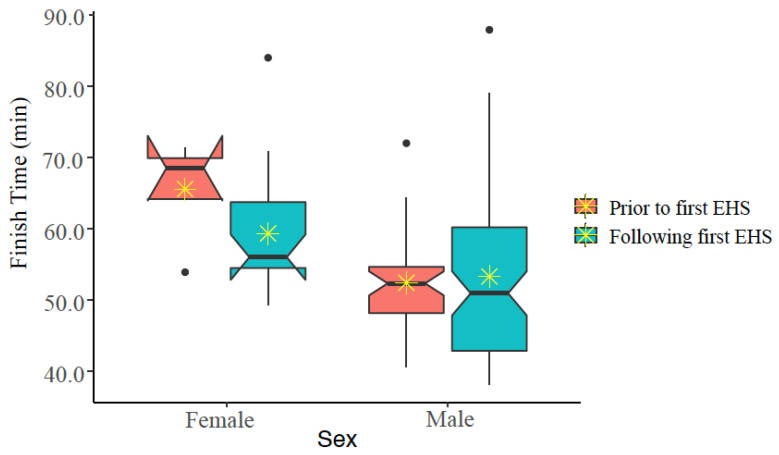
Notched box plot of finish times prior to and following first exertional heat stroke (EHS) episode at the Falmouth Road Race for those having more than one EHS at this race. The box represents the interquartile range, spanning the 25 to 75 percentiles of the data. * indicates mean of the data. The line shows the median of the data, and the ends of each notch represent the 95% confidence interval of the median. Notches on the boxplots overlap for males, but not females, suggesting that FT are not statistically significantly different for males, but that FT is significantly different for females (at alpha ≤ 0.050).

**Figure 7 medicina-56-00720-f007:**
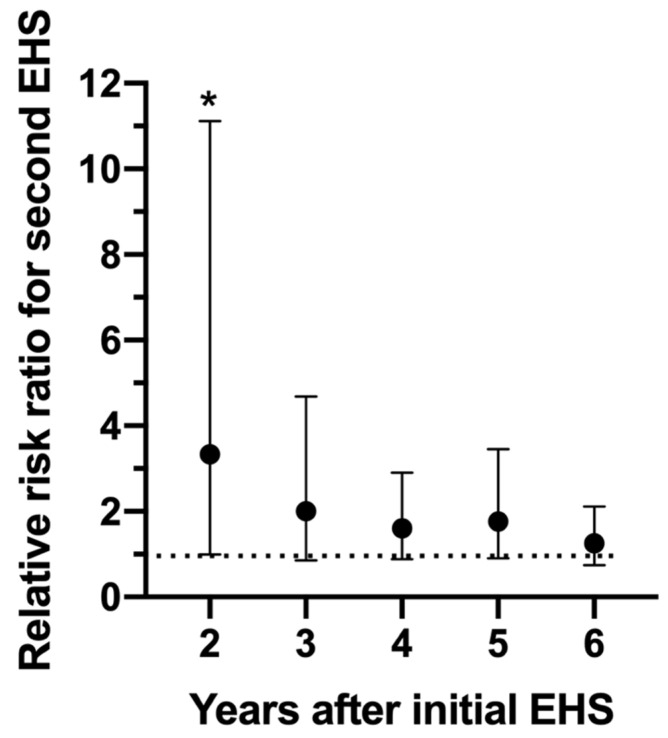
Relative risk ratio of having a second exertional heat stroke (EHS) at the Falmouth Road Race between 2–6 years following the initial EHS episode. * indicates significantly elevated (*p* = 0.050) risk compared to those who only experienced 1 EHS. Bars indicate 95% confidence intervals. Dotted line represents relative risk ratio of 1.0, indicating no difference in risk.

**Figure 8 medicina-56-00720-f008:**
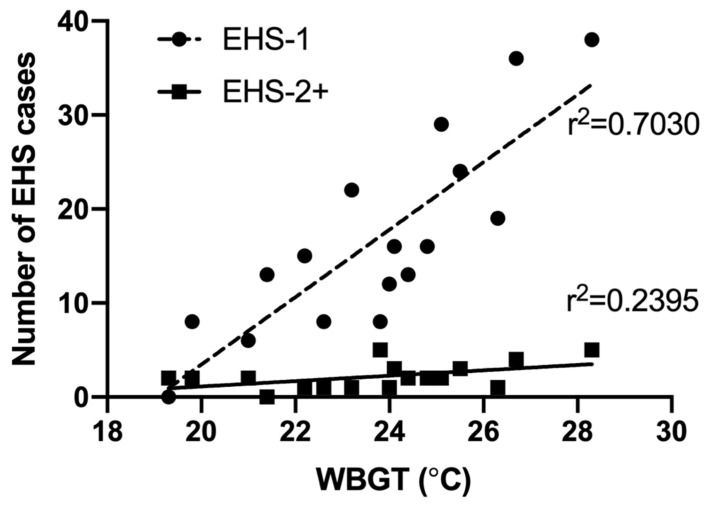
Correlation of race day maximum wet bulb globe temperature (WBGT) and number of exertional heat stroke (EHS) patients seen at the Falmouth Road Race medical tent between 2003–2019 for those with 1 EHS (EHS-1, n = 283 total observations), and those who had a prior history EHS at the same race (EHS-2+, n = 37 total observations).

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
