# Peer review of "Incidence of Recurrent Exertional Heat Stroke in a Warm-Weather Road Race"

_medicina, 2020, doi:10.3390/medicina56120720_

Round 1
Reviewer 1 Report
Introduction:
Lines 53-54: can you rephrase this sentence to highlight the LACK of published case studies? The way it is currently stated does not necessarily emphasize that point.
Line 55: can you provide references to the “handful of case reports”?
Line 65: yet to be supported? Or reported?
Line 80: I think this can be reworded to stress that this actually can be systematically examined, it is just not often feasible (due to inadequate medical personnel, lack of thorough PMHx, etc.).
Methods:
Line 95: can you provide some examples of specific S/S related to the neuropsychological dysfunction?
Line 102: for quantifying the number of cases, if a subject was deemed to have more than 1 case of EHS, was this only specific to the diagnosed cases at this race? Or if they reported that they previously had a case of EHS that was not affiliated with the FRR was that also included?
Lines 110-112: can you speak to the accuracy of this meteorological data? It seems like it may not accurately reflect the conditions of the race since the base is almost 20km away from the location of the race. Also, would it be expected to have different conditions (i.e. differences in windspeed, etc.) at different points along the race route?
Results:
To my above point, it seems like the EHS-2+ group were only identified based on EHS cases at this particular race. Do you think the number of individuals who have experienced multiple cases is actually higher (i.e. if they have suffered an additional EHS outside of this race)?
Line 164: can you present this as % of the first finish time? It may provide some additional context related to overall performance instead of just raw finish time. This could potentially change the appearance of Figure 2 also.
Figure 3: I’m not sure how much this figure contributes to the overall paper. It is somewhat difficult to read and the data may be able to be presented more clearly in an alternative format.
Figure 6: I think this figure is extremely important and of extreme clinical importance. However, I do think it is important to emphasize (probably in the discussion section) that this relative risk is solely based on the amount of time between subsequent EHS cases, and omits other potential factors that could (and probably do) contribute to relative risk.
Figure 7: while statistically significant (barely =)), the correlation for the EHS-2+ is quite low.
Discussion:
Lines 268-271: I think this is an interesting point to discuss but it should be emphasized (maybe in the limitations or future research sections) that other factors that were not considered in the analysis could contribute (i.e. age, fitness level, other individual characteristics, etc.)
Line 274: again, this relationship is extremely weak, despite showing statistical significance.
Lines 281-287: I think this paragraph highlights the importance/need to extend the analysis (in future studies) beyond the variables that were considered in this study.
Reviewer 2 Report
Thank you for allowing me to review the manuscript titled "Incidence of Recurrent Exertional Heat Stroke in a 3 Warm-Weather Road Race". I commend the authors for their write up and analysis of such a large data set and trying to provide novel findings and an application to the public/runner, medical team and organisers regarding the understanding of EHS recurrence risk.
This is a well-written manuscript and provides key information to the reader/community and advances our knowledge of EHS recurrence, well done.
My key thoughts regarding the paper include:
- Inconsistency in figure presentation
- Inconsistency in reporting statistics within the results section
- Some bold statements aligned to the results within the discussion section, yet do not have statistics or provide clarity/comparative data
- Suggest trying to use comparative data from previous literature within the discussion i.e. from case reports or military populations [references 16,18,32] as it is currently lacking much critique/comparison.
- Too many citations used for the same issue/point, 6 x in L52, and 10 x in L72. Please use references wisely and suggest using other literature within the discussion.
Please also find below a list of considerations for your manuscript with associated line numbers.
General comment:
- Spacing issue between data and units (years/min) throughout
- Consistency in use of “participants” and “individuals” not “subjects”
- A lot of references used within the introduction aligned to “previous history of EHS remains a cited risk factor for future EHS risk [1,4,10,11,15,16]” and “delayed treatment or inferior cooling methods [5,7,9,11,13,20,26–29]” yet, very little reference used in the discussion
- Figure quality is very inconsistent.
- Attention to detail in reporting p values with various decimal places
- Lack of real application of the results until the final few sentences of the conclusion.
Specific comments:
- Abstract = Do not agree with lines 31-33, TRE is not higher, and FT is not slower - this is different to the results section - please amend.
- Use of RRR term here, but not in rest of the paper.
- L48 - Consider expanding on “nearly” with some % data from other studies
- L51 – “factors” implies multiple, could you consider providing the other contributing ones opposed to just EHS history.
- L59 – Consider replacing “latter” with 2nd
- L64 - there appears a double space between “tolerance” and “certainly”
- L68 – Hyphen for “long-held”
- L79 – full stop after references not before
- L86 – add “episode” after 1 EHS
- L86 – Is “New Balance” needed here? Has it been the sponsor/race name for the entire 17 year data collection period? Suggest keep FRR.
- L87 – Consider adding in gender and WGBT factor here, as well as an hypothesis
- L92 – Remove “about”
- L125 – Reference for effect size?
- L125 – Can you include relationship interpretation criteria here as this will help in you discussion.
- L156 – Add “individuals” or “participants” to EHS-2+ here and throughout, also remove “subjects”.
- L159 – Consider removing “in other words”
- L162 – Consider removing “initial”
- L165 – Add FT here, and how many were not available
- L166 – Please check if ES and CI has been abbreviated
- Figure 1 = Remove irrelevant capital letters, make figure larger, marker inconsistency, add EHS-1 and EHS-2+, ensure average line is clearer. Also check consistency between Figure 1 and 2 please – “F” is different colour, outline of figure and marker options.
- Figure 2 = All FT consider min:sec reporting
- L174 – Add individuals next to EHS-2+
- Figure 3 = Remove subject, add colour to help show timeline, separate male and female data?
- L189 – Please ensure there is consistency in participants/individuals, male and females, or men and women.
- L189 – There is no stats here for FT, please add in as well as mean difference perhaps.
- L190 – Capital F for Figure
- Figure 4a and 4b are not joined, so need to be 4 and 5, or can you combine figures together?
- Figure 4a and 4b need to have consistency in decimal places with other reported data, and please extend data/units on y-axis to ensure data can be understood at the end of the axis line – currently data appears above the 120 mins and the reader may not be able to understand what this is.
- L203 – Please reword this sentence, it is difficult to understand. Perhaps replace “done” with repeated, add “individuals’” next to EHS-2+.
- L206 – Stats and mean difference needed for FT, “FT” also needed.
- L220 – Replace “subjects”
- L224 – Replace “n” with sample, replace “subjects”
- L225 – Remove unnecessary capitals to years, post, first.
- Figure 6 – Remove unnecessary capitals, add what dotted line represents in figure title, please ensure the title matches that of the text as it appears you are comparing EHS risk over years (as in line 228), then comparing EHS-2+ risk to EHS-1 (as in figure title) – please clarify.
- As mentioned in figure 6 title, “p≤0.05”, please ensure this is repeated in the methods section, as well as then future consistency with reporting p value of 0.046 = 0.05? As this is something you then make a statement based upon in the discussion.
- L235 – Please revisit the way you report your p values, here it is p=0.05 (2 decimal places), others its 0.046, suggest stick with 1 approach and go with 2 decimal places.
- L240 – Hear your need to provide “low, moderate, strong” relationship with the use of threshold criteria from Pearson analysis that should be in the methods.
- L240 – Please also consider how the reader interprets “Frist EHS” and keep consistency with reporting elsewhere.
- Figure 7 – Please consider the consistency in figure format, style, font, bold, size etc, this varies throughout, most likely due to Excel, GraphPad and R formats.
- L243 – Suggest remove part of sentence from “trend….. to…… exploration” as this is interpretating/discussing and should be left for the discussion (this is also repeated in the discussion anyway).
- L255-261 – Consider removing “which is lower than the 15% of reported EHS French military patients 256 who reported a prior EHS episode[18].” AND “When considering recovery time frame, recent data has reported that biochemical recovery lasts up to 16 days post EHS [36]. Additionally in a mouse model an altered heat shock response has been noted at 30 days post EHS [37], though neither examined future exposure or the occurrence of future EHS episodes.” to further down into the discussion as this section should primarily be linked to your main findings.
- L264 – I find this section very hard to follow as there were no significant differences here as reported above in the results section, so how can we (the reader) really infer that TRE was lower/FT was slower – when you previously report no difference? Please reconsider this entire section/paragraph. Likewise from L164-271 can you/do you have any references/data that support this, as you are now inferring that females with EHS-2+ have “improved” their FT, due to exercise intensity (which you have not provided) and this is what you are suggesting for the shorter recurrence time as opposed to males? Your overuse of “tended”, “perhaps”, “may”, “partially” all within a few sentences here prompts much doubt and perhaps should either be left out, and just discuss your data or offer further research is required. This is a major point I would like to see addressed.
- L273 – Please interpret the “significant correlation” with low/moderate/strong as per missing methods thresholds. Further, the use of three decimal places here (i.e. 0.046) specially here as opposed to other sections above where you have typically used 2 prompts doubt that this is only included to be <0.05. Suggest this needs to be addressed and advise with caution.
- L276 – “splice”? please define/expand, I am not familiar with this term.
- L279 – Recurrent use of 3 decimal places, and also, please define and expand on how this is a trend and what is the application of this finding/suggestion.
- L282 – Need to an “episode” next to EHS.
- L284 – Suggest add “the” before “ability”
- L306 – Suggest re-defining EHS-2+ here, as well as removing “patients”.
- Section 313-319 – Consider relating this study to event organisers and the athlete too, not just medical tents, if you are reporting that a runner who has suffered EHS beforehand, may well be at greater risk within the subsequent year or 2, perhaps the race organisers need to screen athletes who sign up for the race for this information and work with the athlete/medical teams to ensure they are fully prepared, especially if heat stress conditions during the race are higher, as seen in your previous work ranging 18-28°
Reviewer 3 Report
The outlined paper provides an interesting snapshot of recurring EHS during a popular road race and raises some interesting points. The paper is well written and structured and makes good use of the available data. It is of particular interest that the authors have identified the notion of increased risk occurring within the 2 yr timeframe and the importance for event organisers to consider medical history.
Whilst the authors provide a critical interpretation of the available data it is noteworthy that they acknowledge the major limitations with the study i.e. the complete lack of data on the additional factors such as training history, fitness, prior illness etc. As such the value of the paper and the meaningfulness of the data and conclusions are limited and speculative.
Specific comments
Line 50 – should “within the initial 30 minutes collapse” not state within the initial 30 minutes AFTER collapse”
Line 52 – the list of referenced articles could include:
Day, T. K., & Grimshaw, D. (2005). An observational study on the spectrum of heat‐related illness, with a proposal on classification. Journal of the Royal Army Medical Corps, 151, 11–18.
Phinney, L. T., Gardner, J. W., Kark, J. A., & Wenger, B. C. (2001). Long‐term follow‐up after exertional heat illness during recruit training. Medicine and Science in Sports and Exercise, 33, 1443–1448
Line 81: The referenced I referred to on the comment re line 52 are worth considering.
Line 104: The authors state “Data was then de-identified for analysis” yet indicate on line 100 that the approval for the study was given by University’s Review Board and was exempt from consent due to the data being de-identified medical records. Can the authors provide clarity on whether participants were identifiable at any point during the study? As the authors state on lines 104-106 that data such as finish time and names on medical charts were consulted, the participants are clearly identifiable. Does this satisfy the Review Board’s requirements?
Line 244: should this read as THEY trend as opposed to “the trend”?
Line 264: The authors state “EHS-2 +tended to have lower initial TRE (0.3°C)….”. This should be clarified as it might be interpreted as “initial” referring to Tre as the beginning of the race.
Line 272-273: Whilst the correlation might be statistically significant an R2 of 0.23 is low and is not indicative of a strong correlation.
Round 2
Reviewer 2 Report
Many thanks to the authors for taking the time to address each of the considerations, points and/or recommendations raised by the reviewer.
A final few considerations below:
- Major = There is still a lot of/too many references used within the introduction aligned to:
- "especially when CWI is not utilized within the initial 30 minutes following collapse [1,4,6–12]" Line 51
- “previous history of EHS remains a cited risk factor for future EHS risk [1,4,10,11,15,16]” Line 54
- “delayed treatment or inferior cooling methods [5,7,9,11,13,20,26–29]” Line 74, I really do not think there is a need for 10 references here, when number 5 is sufficient.
- Line 155 is the alpha level set to less than, or less than and/or equal to 0.050?
- Effect size and confidence intervals needs to be explaine din the methods section, not results. = 175/176
- As before, please provide exact p values. = 203/204
- Please add in p values for female and male data here = 220/221
- It is still unclear what and where "notches" are in the figure 6 = 222
- You need to correct reporting of alpha level in methods for this to be "significant" p=0.050, so you need to ensure you include less than or equal to 0.050 in stats section= 253
- RRR to 2 decimal places in text but none in figure 7, please correct = 255
- how have you defined "marginally" here? =257
- how have you defined "highly" here? = 261 - please add this into the methods section
- delete comma after "limited" = 283
- please define what "elevated markers" = 288
- add "runners or participants or individuals" next to "EHS-2+" = 293, also in 295, please check through that this group is defined accordingly
- remove capitals for females and males = 299
- what is "marginal" = 307
- This sentence is still quite vague and speculative = 315, why "utmost clinical importance", in what context?
- reword Line 242 - the presence of participants? or presence of EHS and repeated cases?
Reviewer 3 Report
Dear Authors,
Thanks for clarifying the points raised.
Best of luck with your work and further publications.
Author Response
The authors would like to thank the reviewers for their additional insight and feedback. We appreciate your insights and expertise towards improving this paper. Thank you again.